# LLM and Simulation as Bilevel Optimizers: A New Paradigm to Advance Physical Scientific Discovery

**Pingchuan Ma**[1], **Tsun-Hsuan Wang**[1], **Minghao Guo**[1], **Zhiqing Sun**[2],
**Joshua B. Tenenbaum**[1 3 4], **Daniela Rus**[1], **Chuang Gan**[5 6], **Wojciech Matusik**[1]

[1]MIT CSAIL, [2]CMU LTI, [3]MIT BCS, [4]CBMM, [5]UMass Amherst, [6]MIT-IBM Watson AI Lab

## Abstract

Large Language Models have recently gained significant attention in scientific discovery for their extensive knowledge and advanced reasoning capabilities. However, they encounter challenges in effectively simulating observational feedback and grounding it with language to propel advancements in physical scientific discovery. Conversely, human scientists undertake scientific discovery by formulating hypotheses, conducting experiments, and revising theories through observational analysis. Inspired by this, we propose to enhance the knowledge-driven, abstract reasoning abilities of LLMs with the computational strength of simulations. We introduce *Scientific Generative Agent* (SGA), a bilevel optimization framework: LLMs act as knowledgeable and versatile thinkers, proposing scientific hypotheses and reason about discrete components, such as physics equations or molecule structures; meanwhile, simulations function as experimental platforms, providing observational feedback and optimizing via differentiability for continuous parts, such as physical parameters. We conduct extensive experiments to demonstrate our framework's efficacy in constitutive law discovery and molecular design, unveiling novel solutions that differ from conventional human expectations yet remain coherent upon analysis.

## 1 Introduction

Physical science automation aims to accelerate discovery [55]. Key aspects of human scientific process include: iterative hypothesis testing [40], discrete and continuous solution components [55], knowledge exploitation with occasional exploration [58], and universal principles with discipline-specific nuances [44]. LLMs excel as generalist tools with vast knowledge [1], aiding scientific discovery through reasoning and natural language interfaces. However, they lack computational capabilities crucial for physical sciences that requires specific domain knowledge.

To this end, inspired by the overarching philosophy of human scientists, we introduce **Scientific Generative Agent (SGA)**, a bilevel optimization approach wherein the outer-level engages LLMs as knowledgeable and versatile thinkers for generating and revising scientific hypothesis, while the inner-level involves simulations as experimental platforms for providing observational feedback. Overall, our contributions are concluded as:

- We present a generic framework for physical scientific discovery that combines LLMs with physical simulations.
- We propose a bilevel optimization with LLMs for discrete-space search-based optimization and differentiable simulations for continuous-space gradient-based optimization.
- We conduct extensive experiments to demonstrate the effectiveness and generality of the proposed framework in physics law discovery and molecular design.

38th Conference on Neural Information Processing Systems (NeurIPS 2024).

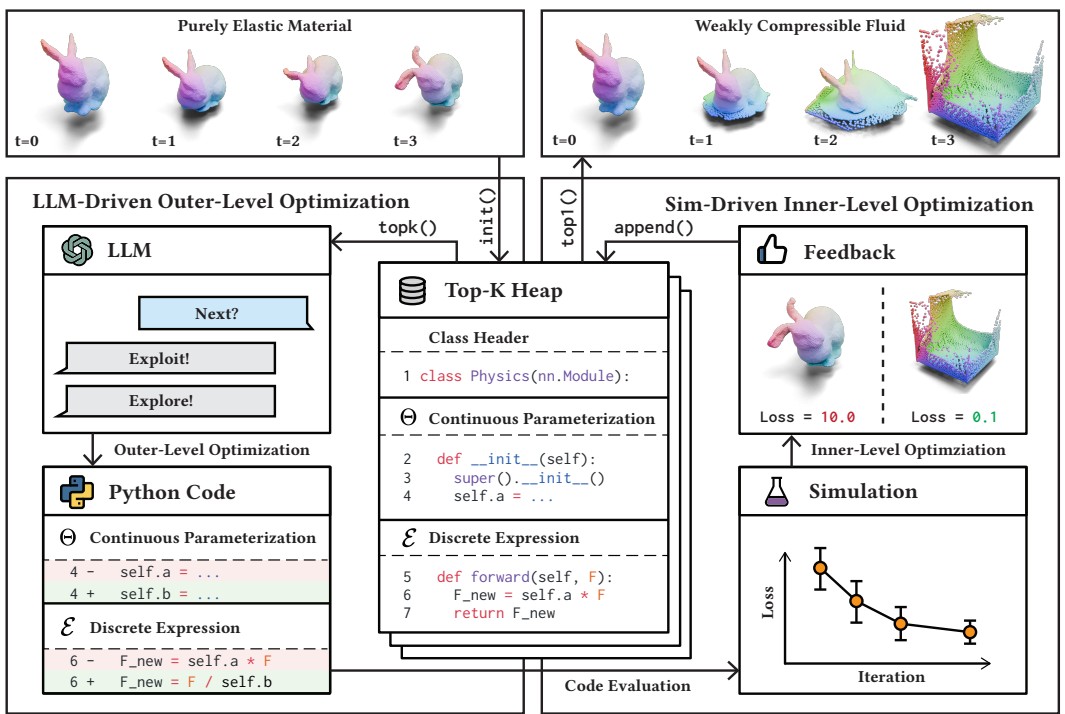

Figure 1: **The overall pipeline of Scientific Generative Agent (SGA).** Taking the constitutive law searching problem as an example, the input is an initial guess (a purely elastic material), and the output is another constitutive law optimized towards the ground truth (weakly compressible fluid).

## 2 Scientific Generative Agent

SGA is a bilevel optimization framework where the upper level features LLMs as proposers of scientific solutions, and the lower level utilizes simulations as experimental platforms for validation. We illustrate the overall pipeline in Fig. 1.

### 2.1 Bilevel Optimization Pipeline

First, we describe the simulation as a process where a simulator takes in a scientific expression and continuous components as inputs and gives simulated physical phenomenon and additional observational feedback as outputs. Next, the LLM acts as a thinker to propose expressions based on past experimental results from simulation. This process involves the LLM taking in a set of past simulation results containing an evaluation of the scientific problem, other physical feedback, and past proposals, along with a prompt. The LLM then outputs proposed expressions and continuous parameterization for the decision variables. With these elements, we define a bilevel optimization problem: the objective is to minimize the evaluation of the simulated physical phenomenon, which depends on the proposed expression, continuous parameterization, and optimal continuous parameters. The optimization problem has two levels: (i) the outer optimization searches for an expression that defines what experiments to be conducted and continuous parameterization that defines the search space of the inner continuous optimization; (ii) the inner optimization, which depends on the outer-level variables, searches for the optimal continuous parameters given the proposed expression via differentiable simulation. We detail the complete algorithm with a python-like pseudo-code in Alg. 1.

### 2.2 LLM-Driven Outer-Level Search

**LLM-driven Optimization**   LLMs are effective for generic optimization through prompting and context [60, 43]. Inspired by [31], we use evolutionary search with multiple offspring per iteration. Our approach selects several high-performing candidates, enhancing hypothesis feasibility and facilitating crossover, with LLMs generating new hypotheses from past experiments [43].

Table 1: **Benchmark**. We use column **#Iter.** as the number of iterations, **#Hist.** as the $K$ value for the top-k retrieval, $\frac{\text{#Exploit}}{\text{#Explore}}$ as the number of offspring for exploitation versus exploration, **Bilevel** as if bilevel optimization is enabled. The best method with the lowest loss is highlighted in **bold** text.

| Method | #Iter. | #Hist. | $\frac{\text{#Exploit}}{\text{#Explore}}$ | Bilevel | Constitutive Law Search | | | | Molecule Design | | | |
|---|---|---|---|---|---|---|---|---|---|---|---|---|
| | | | | | (a) ↓ | (b) ↓ | (c) ↓ | (d) ↓ | (e) ↓ | (f) ↓ | (g) ↓ | (h) ↓ |
| CoT | 1 | 5 | N/A | ✗ | 298.5 | 1462.3 | 150.0 | 384.1 | 3.0 | 32.1 | 18.6 | 6.0 |
| FunSearch | 20 | 2 | 0 / 4 | ✗ | 210.3 | 872.2 | 82.8 | 139.5 | 1.1 | 7.1 | 8.3 | 1.1 |
| Eureka | 5 | 1 | 0 / 16 | ✗ | 128.0 | 531.0 | 101.7 | 150.1 | 4.3 | 9.8 | 3.3 | 9.7e-1 |
| OPRO | 5 | 5 | 0 / 16 | ✗ | 136.2 | 508.3 | 99.2 | 128.8 | 2.4 | 9.4 | 3.1 | 1.3 |
| Ours (no bilevel) | 5 | 5 | 4 / 12 | ✗ | 90.2 | 517.0 | 83.6 | 68.4 | 8.6e-1 | 9.1 | 1.8 | 1.4 |
| Ours (no exploit) | 5 | 5 | 0 / 16 | ✓ | 3.0e-3 | 3.9e-1 | 6.6e-2 | **1.4e-12** | 4.0e-4 | 1.5e-1 | 6.1e-1 | **2.8e-5** |
| Ours | 5 | 5 | 4 / 12 | ✓ | **5.2e-5** | **2.1e-1** | **6.0e-2** | **1.4e-12** | **1.3e-4** | **1.1e-1** | **5.4e-1** | 3.6e-5 |

**Interfacing with Simulation**    Integrating LLMs with simulation requires efficient, structured communication. We use equation searching and entity searching for LLM-to-simulation communication, unified as an abstraction. Equation searching allows LLMs to propose equations and search spaces, while entity searching focuses on structural descriptions. For simulation-to-LLM communication, we use expert knowledge to extract relevant information as feedback, similar to a senior scientist guiding a junior colleague. The inner optimization results also serve as feedback from simulation to LLMs, detailed in the next section.

**Exploitation and Exploration**    We employ an exploit-and-explore strategy by adjusting LLMs' decoding temperature [60], mimicking human scientists' approach to breakthroughs. When generating offspring, we divide them into two groups: cautious followers (exploit) and daring adventurers (explore). We observed that the exploit group often repeats previous solutions, while the explore group tends to yield overly random or invalid solutions. A 1:3 ratio between exploit and explore groups has proven effective emperically based our experiments.

### 2.3    Differentiable Inner-Level Optimization

Inner optimization uses gradient-based methods to find optimal parameters within the search space defined by the outer level. Domain-specific knowledge is distilled through gradients from the simulation to intermediate optimization results. These results, along with the final output, are fed back to LLMs for solution refinement. The feedback may include loss curves and auxiliary recordings, providing information on various aspects of improvement.

## 3    Experiments

### 3.1    Problem Definitions

**Constitutive Law Discovery**    Identifying the constitutive law from motion observations stands as one of the most difficult challenges in fields such as physics, material science, and mechanical engineering. Here we follow the recent advances in physical simulation and formulate the constitutive law discovery task as an optimization problem [29].

**Molecule Design**    We focus on a prevalent task in molecule design: discovering molecules with specific quantum mechanical properties. Our objective is to determine the optimal molecular structure and its 3D conformation to match a predefined target quantum mechanical property. The design process involves both the discrete expression – the molecular structure represented by SMILES strings [57], and the continuous parameters – the 3D coordinates of each atom in the molecule.

### 3.2    Experiment Setup

We design a diverse set of challenging tasks for evaluation. For constitutive law discovery, we propose 4 tasks including: **(a)** fitting the non-linear elastic material starting from a linear elastic material, **(b)** fitting the von Mises plastic material starting from a purely elastic material, **(c)** fitting the granular material starting from a purely elastic material, and **(d)** fitting the weakly compressible fluid starting

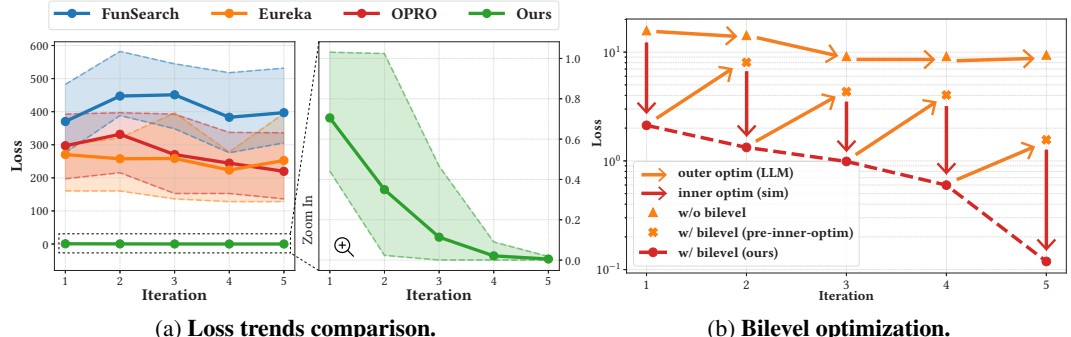

(a) **Loss trends comparison.**     (b) **Bilevel optimization.**

from a purely elastic material. For molecular design task, we consider 4 popular tasks, centering on 3 commonly evaluated quantum mechanical properties [13, 65]: **(e)** HOMO (Highest Occupied Molecular Orbital) set to 0, **(f)** LUMO (Lowest Unoccupied Molecular Orbital) set to 0, **(g)** the HOMO-LUMO energy gap set to 0, and **(h)** the HOMO-LUMO energy gap set to -2.

## 3.3 Physical Scientific Discovery

We consider 6 strong baselines for evaluation: (i) **Chain-of-Thoughts (CoT)** prompting [56] solves the problem by looking at step-by-step solutions from examples. We provide 5 examples with an explanation to CoT as the initial solution. (ii) **FunSearch** [43] utilizes evolutionary strategy to avoid local optimum. We adopt the given hyperparameters from the original implementation with 2 optimization histories and 4 explorers. We set the number of iterations to 20, yielding the same number of solutions evaluated, for a fair comparison to other methods. (iii) **Eureka** [31] generates multiple solutions in each iteration to improve the success rate of the generated code. We keep the hyperparameters from the original implementation. (iv) **Optimization by PROmpting (OPRO)** [60] highlights the advantages of involving a sorted optimization trajectory. We set the hyperparameters to be equal to **Eureka** except for the number of historical optimization steps. In all these works (i-iv), we notice the temperatures for LLM inference are all 1.0, which is equal to the exploring temperature in our method, so we denote them with 0 exploiter. We also consider 2 variants of our method: (v) **Ours (no bilevel)** removes the bilevel optimization by only searching with LLM. (vi) **Ours (no exploit)** removes the exploitation by setting the temperature to 1.0 all the time.

We present our experiments against the 8 designed tasks and show the results in Table 1. Compared to baselines (i-iv), our method is significantly better by a number of magnitudes. When the bilevel optimization is removed from our method, the performance drops dramatically, but still statistically better than baselines (i-iv), indicating the choice of hyperparameters and the integration of exploitation is helpful for the task. When we remove the exploitation but restore the bilevel optimization, we notice the performance grows back. It has comparable performance compared to our method in **(d)** or even better results in **(h)**. However, in some tasks, especially hard ones (e.g., **(b)** and **(f)**) that we care more in reality, the performance gap is over $50\%$, indicating the effectiveness of our exploit-and-explore strategy. We also present the loss trend in task **(a)** in Figure 2a, our method outstands with a much lower loss and a converging trend. We present more experiments in Sec. C.

## 3.4 Bilevel Optimization

Here we evaluate the importance of bilevel optimization in Figure 2b using the task **(h)**. Comparing the blue triangle curve and the red dot curve, which represent the LLM-driven outer-level optimization and the simulation-driven inner-level optimization, it is easy to conclude that the loss performance with bilevel optimization is better. Nevertheless, we are also interested in how bilevel optimization works inside each optimization step and how much LLMs and simulations help respectively. As shown as a zigzag curve, we found that LLMs and simulations help each other over all optimization steps: the next proposal from LLMs will be better with simulation-optimized results, and vice versa. We argue that LLMs and simulations have different expertise: LLMs are generalist scientists who have cross-discipline knowledge, while simulations are domain experts who have specialized knowledge.

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

# A  Related Work

## A.1  Automated Scientific Discovery

Automated scientific discovery, enhanced by machine learning methods, serves as a powerful accelerator for research, enabling scientists to generate hypotheses, design experiments, interpret vast datasets, and unearth insights that may elude traditional scientific methodologies [1, 22, 55]. This multifaceted process unfolds through two synergistically linked stages: hypothesis formation and the collection and analysis of experimental data. The integration of automated systems not only augments the scientific inquiry process but also streamlines the discovery pipeline, from conceptualization to empirical validation. This paper places a particular emphasis on, but is not limited to, constitutive

**Algorithm 1** Scientific Generative Agent

**Require:** Discrete expression and continuous param $(\mathcal{E}, \theta \in \Theta)$, Num of exploiting $M_l$, Num of exploring $M_h$, Exploiting temperature $T_l$, Exploring temperature $T_h$

```
1:  # Store ranked (solution,param) by heap
2:  H ← heap()
3:  # Continuous optimization
4:  θ̂ ← optim(E,θ;Φ)
5:  H.append((E,θ̂))
6:  for i = 1,...,N do
7:      # Generate M_l solutions from LLM
8:      (E,Θ)[:M_l] ← LLM(H.topk(K),T_l)
9:      # Generate M_h solutions from LLM
10:     (E,Θ)[M_l:M_l+M_h] ← LLM(H.topk(K),T_h)
11:     for m = 1,...,M_l + M_h do
12:         # Continuous optimization
13:         θ̂ ← optim(E,θ ∈ Θ;Φ)
14:         H.append((E,θ̂))
15:     end for
16: end for
```

**Ensure:** `H.topk(1)` `# Return the best`

law discovery and molecular design. These areas exemplify the profound impact of automation in unraveling complex material behaviors and in the innovative design of molecules with tailored properties. Automatic identification of constitutive material models has been a long-standing problem and recent works utilizes differentiable simulation [12, 29, 30] to address it as a system identification problem. Leveraging machine learning and artificial intelligence, researchers are able to predict molecular behavior, optimize chemical structures for specific functions, and thus, rapidly accelerate the development of new drugs, materials, and chemicals [17, 66, 46].

### A.2 Large Language Models and Agents

The advancement of Large Language Models (LLMs) such as ChatGPT and GPT-4 has sparked considerable interest in their potential as autonomous agents [6, 35, 36]. Recent developments have shown that LLMs can be enhanced to solve complex problems by creating and utilizing their own tools, as demonstrated in the LATM framework [49], and by acting as optimizers in the absence of gradients, as seen in the OPRO methodology [60]. These approaches signify a shift towards more independent and versatile LLM-based agents capable of generating solutions through self-crafted tools and optimization techniques [8, 62, 61], showcasing their evolving problem-solving capabilities. In the realm of scientific discovery, LLMs have begun to make significant contributions, particularly in mathematics and computational problems. The FunSearch method [43] pairs LLMs with evaluators to exceed known results in extremal combinatorics and online bin packing, illustrating LLMs' ability to discover new solutions to established problems. Similarly, AlphaGeometry's success [50] in solving olympiad-level geometry problems without human demonstrations highlights the potential of LLMs in automating complex reasoning tasks. These examples underline the transformative impact of LLMs in pushing the boundaries of scientific inquiry and automated reasoning.

### A.3 Bilevel Optimization

Bilevel optimization involves a hierarchical structure with two levels of optimization problems, where the solution to the upper-level problem is contingent upon the outcome of the lower-level problem [11]. Bilevel optimization problems are inherently more complex than their single-level counterparts due to the nested nature of the optimization tasks and the intricate interdependencies between them. Recent advancements have focused on developing efficient algorithms, including evolutionary algorithms [48], gradient-based approaches [28], and approximation techniques [47], to tackle the computational challenges presented by the non-convex and non-differentiable characteristics of many bilevel problems. Among a wide span of application domains of bilevel optimization, neural architecture search (NAS) [27, 4, 7, 59] is prominent and close to the problem setting in this paper: the upper level optimizes the discrete neural network architecture while the lower level optimizes

Table 2: **Symbolic Regression.**

| Method | R2 ↑ | MSE ↓ | MAE ↓ | Symbolic |
|--------|------|-------|-------|----------|
| **AIFeynman** [51] | 0.05105 | 22814675.8 | 2520.0 | ✓ |
| **DSR** [39] | 0.57527 | 10966411.0 | 2045.0 | ✓ |
| **BSR** [18] | 0.66526 | 8642965.0 | 1938.6 | ✓ |
| **AdaBoost** [45] | 0.75058 | 6439962.9 | 1777.7 | ✗ |
| **GP-GOMEA** [54] | 0.77734 | 5749076.4 | 1580.1 | ✓ |
| **SBP-GP** [53] | 0.81773 | 4706077.0 | 1367.5 | ✓ |
| **LightGBM** [19] | 0.83368 | 4294433.7 | 1129.9 | ✗ |
| **XGBoost** [10] | 0.87775 | 3156500.5 | 1109.2 | ✗ |
| **MRGP** [3] | 0.91074 | 2304682.5 | 950.5 | ✓ |
| **EPLEX** [23] | 0.91851 | 2104070.1 | 122.2 | ✓ |
| **FFX** [33] | 0.93124 | 1775263.7 | 801.7 | ✓ |
| **MLP** | 0.98240 | 454461.5 | 366.3 | ✗ |
| **FEAT** [9] | 0.98761 | 319800.6 | 336.1 | ✓ |
| **DSO** [34] | 0.99642 | 92374.9 | 168.6 | ✓ |
| **Operon** [21] | 0.99684 | 81577.9 | 92.4 | ✓ |
| **SymbolicGPT** [52] | 0.52333 | 6862154.7 | 1680.7 | ✓ |
| **NeSymReS** [5] | N/A to >3 variables | | | ✓ |
| **T-JSL** [26] | N/A to >2 variables | | | ✓ |
| **Ours** | **0.99901** | **17424.6** | **86.4** | ✓ |

the continuous weights of the neural network. However, typical NAS methods require a predefined search space, constraining the exploration of discrete network architectures to manually specified boundaries. Our framework distinguishes itself by employing LLM encoded with general knowledge and gets rid of the limitations imposed by manual design constraints.

## B   More Explanations

### B.1   Implementation Details

We run all our experiments 5 times with different random seeds following previous practices [31]. Due to the complexity of the task, we provide a simple bootstrapping example of a valid design to ensure the success rate. We use warp [32] for the differentiable MPM simulation, and we develop our inner-level optimization upon PyTorch [38]. In all our experiments, we use mean square error as the criteria and Adam optimizer [20]. We choose `gpt-4-turbo-preview` as the backbone model for LLM and tentatively set the exploiting temperature $T_l = 0.5$ and exploring temperature $T_h = 1.0$.

For the generation of 3D conformations, we utilize the ETKGD algorithm [42] followed by optimization using the Merck Molecular Force Field (MMFF) [14], both implemented within the RDKit [25]. To get the quantum mechanical property values, we employ UniMol [65], a pre-trained transformer-based large model, which has been fine-tuned on the QM9 dataset [41].

### B.2   Algorithm

We attach the full python-like pseudo-code of Scientific Generative Agent pipeline in Alg. 1.

### B.3   Data Workflow

The full input to LLM has 3 main parts: (i) system prompt, (ii) iteration information, and (iii) format prompt. For the system prompt, we insert it into the LLM at the beginning or input it as a special instruction depending on the type of LLM. For the iteration information, we first concatenate the code and its feedback and then simply stack the top $K$ solutions. Finally, we append the format prompt at the end of the prompt to regularize the expected output. From our experiments, it is important to keep the order of prompts to ensure the performance and the successful parsing. More precisely, we show this process in the following python-like code:

Table 3: **Comparison with population-based molecule design**.

| Method | (e) $\downarrow$ | (f) $\downarrow$ | (g) $\downarrow$ | (h) $\downarrow$ |
|---|---|---|---|---|
| **GhemGE** | 4.8e-3 | 1.8 | 1.5 | 9.8e-5 |
| **Ours** | **1.3e-4** | **1.1e-1** | **5.4e-1** | **3.6e-5** |

Table 4: **Experiment in imaginary constitutive law**.

| Method | FunSearch | Eureka | OPRO | Ours |
|---|---|---|---|---|
| **Loss** | 105.0 | 89.1 | 98.0 | **1.3e-3** |

```
1  prompts = []
2  prompts.append(system_prompt)
3  for solution in reversed(solutions.topk()):
4      iteration_prompt = solution.code + '\n' + solution.feedback
5      prompts.append(iteration_prompt)
6  prompts.append(format_prompt)
7  full_prompt = '\n'.join(prompts)
```

### B.4 Differences to Symbolic Regression Task

- Our problem focuses on loss-guided general scientific discovery, which is a super-set of regular regression problems. In the constitutive law search tasks, we do not directly feed the input/output pair to our method. Instead, we consider a much more challenging task: apply the generated constitutive law recursively and use the overall loss as the performance metric. Concretely, a classic SR methods solve $\arg\min_f \|f(X) - y\|$ given $< X, y >$ pairs, whereas our method solves $\arg\min_f \|g(f(X))\|$ given $< X, g(f(X)) >$ pairs and $g$ is a complex function like physical simulation. It is easy to construct $g$ to cover the former case using the later formulation, proving the generality of our problem setup. We formulate our problem as such to reflect a more realistic scenario in scientific discovery, where direct supervision is extremely sparse.

- Our method supports arbitrary number of input variables and output features, where most of SR methods [52] have limitation on the number of input and output. The input limitation strongly caps the complexity of tasks they can solve, and the output limitation forces them ignore the structural correlation between each output dimension. In a comparison, our method supports arbitrary problem settings thanks to the code-based representation, which enables multi-dimensional arrays and tensor operations.

- Our model adapts to multi-discipline application easily, while traditional SR methods typically incorporate with domain-experts' priors via hard-coded constraints and heuristic [51], which is limited, domain-specific, and difficult to customize. Our method is built upon LLMs pre-trained on internet-level data that contains multi-discipline natural languages, mathematical expressions, and codes. As a result, it is easy for users to customize it and adapt to their own diverse applications via natural language guidance.

## C More Experiments

### C.1 Symbolic Regression

We also compare our method with traditional methods in each specific area to demonstrate the generalizability of our method. First, we reformulate our constitutive law search task **(a)** into a symbolic regression task by (i) capture the ground-truth output (the stress tensors) as the supervision, and (ii) separate the 9 output dimension into 9 independent problems and ensemble them for evaluation. Note that these modifications dramatically simplified the original task: we removed back-propagation through time (BPTT) and directly discover the constitutive law without surrogate loss. We evaluate 14 traditional baselines in SRBench [24] and 3 data-driven pre-trained baselines. As shown in Table 2, our method topped on this task even with a much more challenging setting. Also, since our method depends on the in-context learning ability of LLMs, it has little constraint in the number of variables than the data-driven pre-trained baselines.

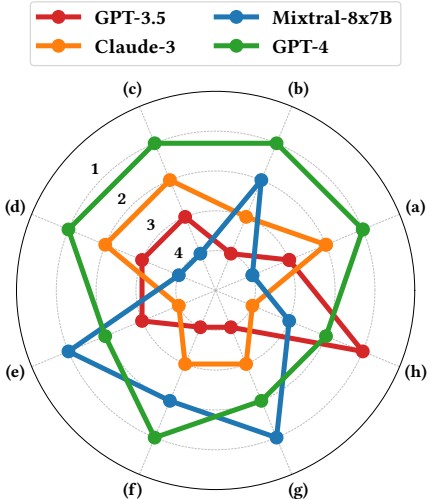

Figure 3: **Backbone LLM.**

## C.2 Population-based Molecule Design

For molecule design tasks, we also compare our method with GhemGE [63], which employs a population-based molecule design algorithm. As shown in Table 3, our method presents a much lower loss, demonstrating the general effectiveness of our method.

## C.3 Generalization or Memorization

In order to figure out if the improvement introduced by our method is merely because the LLM saw the solutions in its training phase, we design an experiment ablating it by making it invent an imaginary constitutive law that does not exist on the earth. We mix the constitutive law of von Mises plasticity, granular material, and weakly compressible fluid by 50%, 30%, and 20%, so that the new constitutive law represents an imaginary material whose behavior is extremely complex. We repeat our experiment setup as in Figure 1. We compare our method against the baselines and report the performances in Table 4. As shown in the table, our method can still discover the constitutive law with a low quantitative loss. From our observation, there is *very little visual difference* between the ground-truth material and the optimized constitutive law.

## C.4 LLM Backbone

In addition to GPT-4 [36], we repeat the experiments in Table 1 using 3 additional LLM backbones: (i) GPT-3.5 [37], (ii) Claude-3-Sonnet [2], and (iii) Mixtral-8x7B [16], and report the rank of them in Figure 3. Indicated by the largest area, GPT-4, as our choice, statistically outperforms the other methods. Interestingly, we found Claude-3-Sonnet is the second top method on most of constitutive law search task, while Mixtral-8x7B even tops on 2 molecule design tasks. As a result, our workflow also works for other LLMs, however, our suggestion for practitioners is to try GPT-4 as the first choice but also consider open-source model (e.g., Mixtral-8x7B) for budget or customizability.

## C.5 Exploitation v.s. Exploration

We visualize the statistics of the simulation execution status in Figure 4 (a) using the task **(b)**, which is one of the most challenging tasks in our experiments. When the exploitation is removed, the error rate dramatically increases, as shown by a decrease in green bars. It leads to a degeneration in the performance of the methods with exploitation as shown in Figure 4 (b). However, even though the success rate remains high, when exploration is removed, the optimization result is still worse than keeping them both. We argue that exploration is significant when the optimization problem is challenging, especially in our case, where the search space is highly non-linear and unstructured and resulting in numerous local optimum.

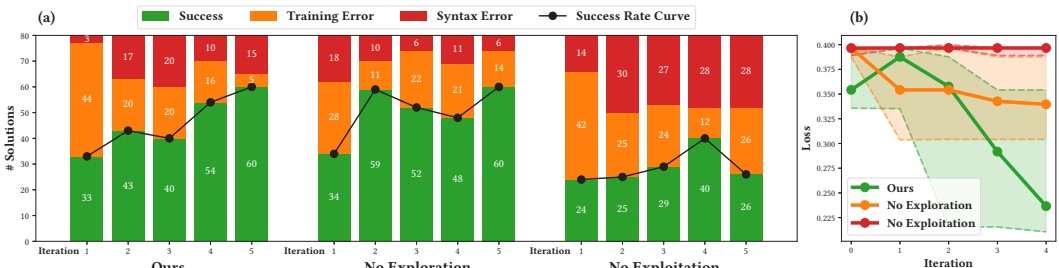

Figure 4: **Exploitation v.s. Exploration.**

Table 5: **Longer Iteration.**

| #Iterations | (a) ↓ | (b) ↓ | (c) ↓ | (d) ↓ | (e) ↓ | (f) ↓ | (g) ↓ | (h) ↓ |
|---|---|---|---|---|---|---|---|---|
| **5** | 5.2e-5 | 2.1e-1 | 6.0e-2 | **1.4e-12** | **1.3e-4** | 1.1e-1 | 5.4e-1 | 3.6e-5 |
| **20** | **4.2e-6** | **4.0e-4** | **2.5e-3** | **1.4e-12** | **1.3e-4** | **6.5e-2** | **1.2e-1** | **5.6e-6** |
| **Improvement** | +1138.1% | +52400.0% | +2300.0% | 0.0% | 0.0% | +69.2% | +350.0% | +542.9% |

## C.6 Longer Iteration

In order to further investigate the potential of our method and ablate the hyper-parameters for practitioners, we add a new study in terms of the number of iterations (question-answering cycles). We repeat our experiment in Table 1 with a prolonged number of iterations to 20 and report the performance in Table 5.

As shown in the table, the number of iterations turns out to be a determining hyper-parameter with significant impart on the performance. While it has little affect on relatively easier tasks, it dramatically improves the performance of the most challenging tasks including **(b)** and **(c)**. For practitioners, the number of iteration should be first considered as the most important hyper-parameter when adapting our method to their own tasks.

## D Case Study

### D.1 Constitutive Law Search

We provide a trimmed snippet of our searched constitutive law in Figure 5 (a) for task **(a)** where a highly non-linear material is provided as the trajectory to fit. We reformat the code slightly to fit into the text. Starting from a linear material, our method is able to automatically generate the constitutive law with a quadratic deviatoric term. Note that our method also provides a concrete implementation of `__init__` function that defines the continuous parameters in the computational graph for later inner-level optimization.

### D.2 Molecule Design

When comparing the two molecules with respect to their HOMO-LUMO energy gap based on optimized results from the LLM as shown in Figure 5 (b), we observe distinct characteristics in each: (i) **Molecule A** (gap-0) includes sulfur and chlorine atoms attached to a ring, coupled with a trifluoromethyl group, introducing electron-withdrawing effects, and (ii) **Molecule B** (gap-2) includes oxygen (notably in ethers) and sulfur within the ring structures introducing localized non-bonding electron pairs. Furthermore, the overall structure of Molecule B is more complex than that of Molecule A, containing multiple rings. An intriguing aspect of Molecule B, which might initially defy expectations, is the presence of a single fluorine atom. The high electronegativity of fluorine typically leads to electron density withdrawal, influencing the gap value. However, due to the complexity of Molecule B's structure, the impact of the fluorine atom is somewhat localized, thereby not significantly altering the gap value.

**(a)**

```python
class Physics(nn.Module):
    def __init__(self, youngs_modulus_log: float = 13.03,
                       poissons_ratio_sigmoid: float = -1.99):
        super().__init__()
        self.youngs_modulus_log = nn.Parameter(
            torch.tensor(youngs_modulus_log))  # Log of Young's modulus
        self.poissons_ratio_sigmoid = nn.Parameter(
            torch.tensor(poissons_ratio_sigmoid))  # Sigmoid of Poisson's ratio
    def forward(self, F: torch.Tensor) -> torch.Tensor:
        youngs_modulus = self.youngs_modulus_log.exp()
        poissons_ratio = torch.sigmoid(self.poissons_ratio_sigmoid) * 0.49
        mu = youngs_modulus / (2 * (1 + poissons_ratio))  # Shear modulus
        lam = youngs_modulus * poissons_ratio / (
            (1 + poissons_ratio) * (1 - 2 * poissons_ratio))
        # Deformation gradient determinant J
        J = F.det().view(-1, 1, 1)
        F_invT = F.inverse().transpose(1, 2)
        # Volumetric part
        P_vol = lam * (J - 1) * F_invT
        # Deviatoric part
        P_dev = mu * (F - (1 / J) * F_invT)
        # Compute Kirchhoff stress tensor
        kirchhoff_stress = P_vol + P_dev @ F.transpose(1, 2)
        return kirchhoff_stress
```

**(b)**

| Molecule A | Molecule B |
|---|---|

C1CC(SC1Cl)C(C(F)(F)F)N      C1OC2SC3C4OC(F)S4C13C2

Figure 5: **Case Study.**

## E    Conclusion and Limitations

We consider a few limitations and future directions. (i) Although we prompt the LLM to generate pseudo-code plans and comments, it is generally hard to ensure the interpretability of LLM-generated solutions. (ii) Since the LLM-generated codes are executed directly without any filtering in our application, there exists potential AI safety risk that hazards the operating system. (iii) Our method only utilizes the internal knowledge of LLMs as the prior, where in reality people design manual constraints and rule to regularize and improve the optimization [51]. We leave these domain-specific applications and human feedback-based regularization methods as our future work. (iv) The performance our method highly depends on the differentiablity of the generated code. However, Zero-order optimizers [15] should also shine since the number of continuous parameters is relatively limited. (v) LLM inference requires large computational resources and thus increases expense. For example, it spends around \$10 for our method to complete one task using GPT-4, which will be increasingly inacceptable when the number of iteration grows. (vi) Due to the reuse of previously generated solutions in our proposed top-k heap, the KV cache in LLM will be highly similar between neighbor iterations. It opens a gate for recent KV cache optimization methods [64] to speedup our method by KV cache reusing.

In conclusion, we present Scientific Generative Agent, a bi-level optimization framework: LLMs serve as knowledgeable and adaptable thinkers, formulating scientific solutions like physics equations or molecule structures; concurrently, simulations operate as platforms for experimentation, offering observational feedback and optimizing continuous components like physical parameters. We focused on two scientific problems: constitutive law search and molecular design. Our approach outperforms other LLM-based benchmark methods, delivering consistent, robust, and nearly monotonic improvement. Furthermore, it shows exceptional ability in identifying unknown, true constitutive laws and molecular structures. Remarkably, our system generates innovative solutions that, despite being unconventional, are deemed reasonable after being thoroughly analyzed by experts in their respective domains. We view our process as a trailblazer, establishing a new paradigm for utilizing LLMs and simulations as bilevel optimization to further advancements in physical scientific discoveries.

