# OpenReview forum: "LLM and Simulation as Bilevel Optimizers: A New Paradigm to Advance Physical Scientific Discovery"
_NeurIPS.cc/2024/Workshop/MATH-AI — MATH-AI 24_

### Official Review · Reviewer_k6gN · 2024-10-05
**Strong Methodology in Physical Scientific Discovery**

**Rating:** 8
**Confidence:** 3

**Review:**

**Summary**

The paper presents the Scientific Generative Agent (SGA), a bilevel optimization framework that combines the abstract reasoning capabilities of Large Language Models (LLMs) with the computational strength of simulations to enhance physical scientific discovery. In this framework, LLMs are used to propose hypotheses in a discrete optimization setting, while simulations perform continuous optimization through experimental feedback. The authors conduct extensive experiments demonstrating the efficacy of the proposed method in the domains of constitutive law discovery and molecular design, which reveals novel solutions that challenge conventional approaches.

**Strengths**
1. The proposed SGA framework is innovative. It combines LLM-based hypothesis generation with simulations in a bilevel optimization framework.
2. The paper demonstrates that the framework is applicable to multiple challenging domains like molecular design and constitutive law discovery.
3. The experiments are well-executed, and the results demonstrate significant improvements over several strong baseline methods.


**Weaknesses**
1. The caption for Figure 2 is missing. Also, it will be helpful to expand the descriptions for Figures 3-5 to clarify the specific tasks they illustrate and the motivations behind them.

2. There is limited discussion regarding the interpretability of the solutions generated by the LLM.

3. The paper briefly mentions the computational requirements, but more detailed information on the computational costs associated with using LLMs and simulations together would be helpful, especially for practitioners considering practical deployment.

**Questions**

How well does the proposed bi-level framework scale with increased problem complexity, especially considering larger simulation environments or more complex LLM hypotheses?

---

### Official Review · Reviewer_L44b · 2024-10-06
**Review of LLM and Simulation as Bilevel Optimizers: A New Paradigm to Advance Physical Scientific Discovery**

**Rating:** 8
**Confidence:** 3

**Review:**

This paper introduces a scientific generative agent framework for advancing scientific discovery---this framework builds on a bilevel optimization framework, which simultaneously optimizes the machine learning model for generating hypothesis as well as the evaluation of past experimental results from simulation. This framework is tested on new applications, namely molecular design and constitutive law discovery.

This is a strong submission---the arguments are convincing, and the framework is very promising. I particularly enjoyed reading the related work and the thoughtfulness in writing, including discussing recent trends in agent-based learning. I recommend accepting this paper as an oral presentation if space permits.

More detailed notes:
- In Table 4, the presented results are orders better than the baselines---perhaps worth explaining this discrepancy in the text.

- This framework reminds me of GAN, since the LLM is essentially treated as a generator, so it may be worth discussing that literature in the related work.

---

### Official Review · Reviewer_Eo73 · 2024-10-08
**Interesting Paper on the Integration of LLMs and Simulations**

**Rating:** 7
**Confidence:** 4

**Review:**

This paper introduces a bilevel optimization framework that combines LLMs with physical simulations to advance scientific discovery.  In their framework, LLMs are used to propose scientific hypotheses and reason about discrete components, while simulations are used to provide observational feedback and optimize continuous parameters. The authors demonstrate effectiveness in two scientific tasks.

**Strengths:**
- I really enjoyed reading this paper. The direction of combining hypothesis generation capabilities of LLMs with Verifiability from scientific simulations sounds interesting for scientific discovery.
- The authors conduct extensive experiments, including ablation studies and comparisons with methods in specific domains.

**Comments and Questions**
- How does the computational time of SGA compare to traditional methods in each domain? This information would help readers better understand the practical tradeoffs of using this approach.
- Have the authors considered ways to evaluate the interpretability and novelty of the LLM-generated hypotheses compared to the knowledge from the literature? This could be crucial for adoption in scientific communities and future research.
- I'm wondering how this framework deal with situations where simulation results contradict its proposed hypotheses? Is there a mechanism for analyzing such conflicts or these solutions are simply ignored?

---

### Decision · Program_Chairs · 2024-10-08

Accept